# The role of meteorological factors on mumps incidence among children in Guangzhou, Southern China

**Jianyun Lu[1]☯, Zhicong Yang[2]☯, Xiaowei Ma[3]☯, Mengmeng Ma[1], Zhoubin Zhang[4]\***

**1** Department of Infectious Disease Control and Prevention, Guangzhou Center for Disease Control and Prevention, Guangzhou, Guangdong Province, China, **2** Director, Guangzhou Center for Disease Control and Prevention, Guangzhou, Guangdong Province, China, **3** Department of Public Health Emergency Preparedness and Response, Guangzhou Center for Disease Control and Prevention, Guangzhou, Guangdong Province, China, **4** Vice Director, Guangzhou Center for Disease Control and Prevention, Guangzhou, Guangdong Province, China

☯ These authors contributed equally to this work.
\* zhoubinzhanggz@21cn.com

**Data Availability Statement:** All relevant data are within the manuscript and its Supporting Information files.

**Funding:** This study was supported by the Medical Health Technology Project for Guangzhou

## Abstract

Mumps, a common childhood disease, has a high incidence in Guangzhou city, China. It has been proven that mumps is influenced by seasonality. However, the role of meteorological factors among children is yet to be fully ascertained. This study explored the association between meteorological factors and the incidence of mumps among children in Guangzhou. Distributed lag nonlinear models were used to evaluate the correlation between meteorological factors and the incidence of mumps among children from 2014–2018. The nonlinear lag effects of some meteorological factors were detected. Mean temperature, atmospheric pressure, and relative humidity were positively correlated with mumps incidence, contrary to that of wind speed. Extreme effects of temperature, wind speed, atmospheric pressure, and relative humidity on the incidence of mumps among children in Guangzhou were evaluated in a subgroup analysis according to gender and age. Our preliminary results offered fundamental information to better understand the epidemic trends of mumps among children to develop an early warning system, and strengthen the intervention and prevention of mumps.

## Introduction

Mumps is a common childhood disease caused by the mumps virus that mostly affects children aged 5–9 years [1, 2]. It manifests clinically as an inflammation of the parotid gland with precursory fever. Most cases are mild and self-limiting, but can result in some severe complications such as deafness, aseptic meningitis, and encephalitis [3]. The incubation period of mumps is 12–24 days, with a median of 19 days [4]. Mumps is not only transmitted by droplet spread, but also via direct contact or contaminated fomites [1].

Mumps can be effectively prevented by vaccination. A sharp reduction in incidence was observed after routine vaccination [5]; however, many countries, including France [6], Ireland

(20181A011051) and the Science and Technology Project of Guangzhou (201804010093) and the Project for Key Medicine Discipline Construction of Guangzhou Municipality (2017-2019-07).

**Competing interests:** There is no conflict of interest for the authors.

[7], UK [8], and USA [9] have experienced resurgences of mumps and reported large scale outbreaks over the past decade. Since 2007, the mumps vaccine has been integrated into the National Immunization Program in China, and the measles, mumps, and rubella (MMR) or measles and mumps (MM) vaccine has been provided free of charge to children aged 18–24 months in Guangzhou since 2008 [10]. However, the incidence rates of mumps in China rose to 33.9/100,000 and 35.6/ 100,000 during 2011–2012 [11]; meanwhile, 10008 and 7856 mumps cases were reported in Guangzhou [10]. A one dose vaccine program is unavailable to fully control the epidemic of mumps, although the immunization coverage rate is about 80% to 90% nationwide [12]. The two doses vaccination program was recommended by the World Health Organization [13] and the polyvalent mumps vaccine was recommended in the USA [9]. Hence, mumps is still a serious public health concern among children not only in Guangzhou, China, but also worldwide.

The incidence of mumps showed apparent seasonality in various areas, with the peak in late spring, early summer, and winter. Different seasonal distributions were even observed in southern and northern China [11, 12]. This hinted that mumps incidence was influenced by seasonal undulation. Therefore, we hypothesized that meteorological factors might play a vital role in the spread of mumps. Recently, the effects of meteorological factors on some infectious diseases have been explored extensively as early warning signals of possible epidemics, such as those of hand, foot, and mouth disease [14], varicella [15], and influenza [16]. Limited studies have been conducted to explore the association between meteorological factors and the incidence of mumps. A study using the ARIMAX model in Beijing, China showed that increasing temperature would cause an increase in the incidence of mumps [17], whereas a study that used a generalized additive model in Jining showed that an increasing trend in the incidence of mumps was observed when the temperature rose above 4˚C [18]. However, a study which applied Pearson's correlation analysis in Czech Republic showed that there was no relationship between mumps incidence and drier and warmer weather [19]. Besides, a Poisson regression model was applied in a study on mumps in 10 cities in Guangxi Province [20]. However, most of the previous studies only focused on linear effects but not lag effects. A study conducted in Shenzhen city applied the distributed lag nonlinear model (DLNM) in exploring the relationship between wind velocity and mumps cases [21]. Studies conducted in Fujian Province and Guangzhou city applied the DLNM in exploring the nonlinear correlations between meteorological factors and the incidence of mumps [22, 23]. However, the nonlinear and lag effects of climatic factors on the transmission among children, who are the main population affected by mumps, are still unclear at present. A study conducted in Japan explored the linear relationship between meteorological factors and mumps incidence among children via negative binomial regression analysis [24]. What are the roles of meteorological factors on mumps incidence among children? There is an urgent need to explore these nonlinear relationships using lag effects.

In this study, an ecological methodology was used to analyze the features of mumps in Guangzhou from 2014 to 2018. We built DLNMs [25], which can flexibly depict the nonlinear correlation and describe delayed effects in time-series studies, to analyze the effects of meteorological factors on mumps transmission among children. The findings of our study may be helpful in developing an early warning system for mumps epidemics among children, improving preventative measures, and reducing the risk of mumps infection among children.

## Materials and methods

### Study region

Guangzhou is the third largest city in China with a total of 8.97 million registered inhabitants. It is a typical meteorologically humid subtropical area and is located within 112˚57′E to 114˚

3′E and 22˚26′N to 23˚56′N (**Fig 1**). In Guangzhou, the summer season is wet and hot, whereas the winter season is dry and cool.

## Data sources

In this study, clinically diagnosed cases of mumps among persons younger than 18 years were obtained from the National Notifiable Disease Report System from January 2014 to December 2018. Clinically, mumps was defined as an acute onset of bilateral or unilateral tender, self-limited swelling of the parotid or other salivary glands lasting for more than 2 days. Mumps is a notifiable disease in Guangzhou; consequently, all clinicians are required to compulsorily report cases to local health authorities within 24 hours via the National Notifiable Disease Surveillance System.

Daily meteorological data, including mean temperature (˚C), diurnal temperature range (˚C) (highest temperature minus lowest temperature), atmospheric pressure (hPa), aggregate rainfall intensity (mm), wind speed (m/s), relative humidity (%), and sunshine duration (hours) were collected simultaneously from the Guangzhou Meteorological Bureau.

## Statistical analysis

We performed a descriptive analysis of meteorological variables and the occurrence of mumps in Guangzhou. Based on scatter plots, we concluded that the correlations between the meteorological factors and mumps incidence were nonlinear. The Mann–Kendall and Pettitt tests were applied to initially analyze the tendency of the meteorological variables and mumps cases among children. Most of the variables tended to increase or decrease. Due to the reasons mentioned above, DLNMs were applied to detect the relationships between meteorological variables and the occurrence of mumps. Atmospheric pressure was strongly correlated with average temperature ($r$ = -0.82, $P<0.01$) as per the Spearman's correlation analysis. Hence, the two variables were not combined in a model to avoid the potential problem of collinearity. Poisson regression, performed with the quasi-Poisson function, was used to assess over-

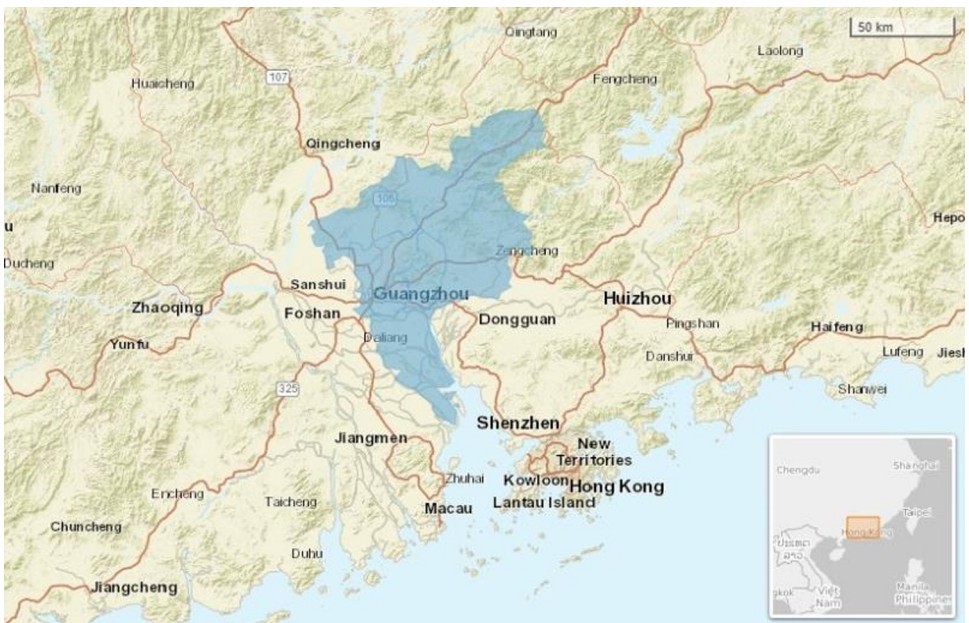

**Fig 1. The geographical location of study area in Guangzhou city, Southern China.**

dispersion of daily cases of mumps in the DLNMs. The calculation is performed using the following equation:

$$\log[\mathrm{E}(Y_t)] = \alpha + \sum \mathrm{NS}_i(X_i, \mathrm{df}_i) + \delta\mathrm{Dow} + \delta\mathrm{Holiday}_t + \mathrm{NS}(\mathrm{Time}, 7 \text{ per year}),$$

where $Y_t$ represents the number of cases of mumps reported on day $t$; both $\alpha$ and $\delta$ are the intercepts; NS represents natural cubic spline; $X$ represents meteorological variables, such as temperature, diurnal temperature range, rainfall, atmospheric pressure, wind speed, relative humidity, and sunshine hours; df is the degrees of freedom; Holiday is defined as a binary variable (if day $t$ is a public holiday or school vacation, it is marked as 1, otherwise it is marked as 0); Dow represents the day of week effect; and Time denotes long term trends and seasonality. We selected the df of day to control long term trends, seasonality, and meteorological variables using the Akaike information criterion (AIC). Based on the AICs, we chose df = 3 for NS according to the meteorological variables mentioned above. After combining the incubation period of mumps, df = 24 was chosen for mean temperature, atmospheric pressure, and wind speed, and df = 14 was chosen for relative humidity.

Extreme high effects were defined as when the value of meteorological variables were beyond the 97.5th percentile; when the value was below the 2.5th percentile, it was defined as an extreme low effect.

All statistical tests were two-sided. Analysis items with $P<0.05$ were considered statistically significant. All statistical analyses were performed using R version 3.5.3 (The R Development Core Team, Vienna, Austria) and the DLNMs were built using the *dlnm* package in the R environment [25].

We performed the sensitivity analyses by shifting the df (5–9) per year to control for seasonality and long-term trends using time, and changing the df (4–7) for mean temperature, atmospheric pressure, diurnal temperature range, wind speed, accumulated rainfall, relative humidity, and sunshine hours.

## Ethical considerations

The study protocol was approved by the Ethics Committee of the Guangzhou Center for Disease Control and Prevention. All data were fully anonymized before we accessed them.

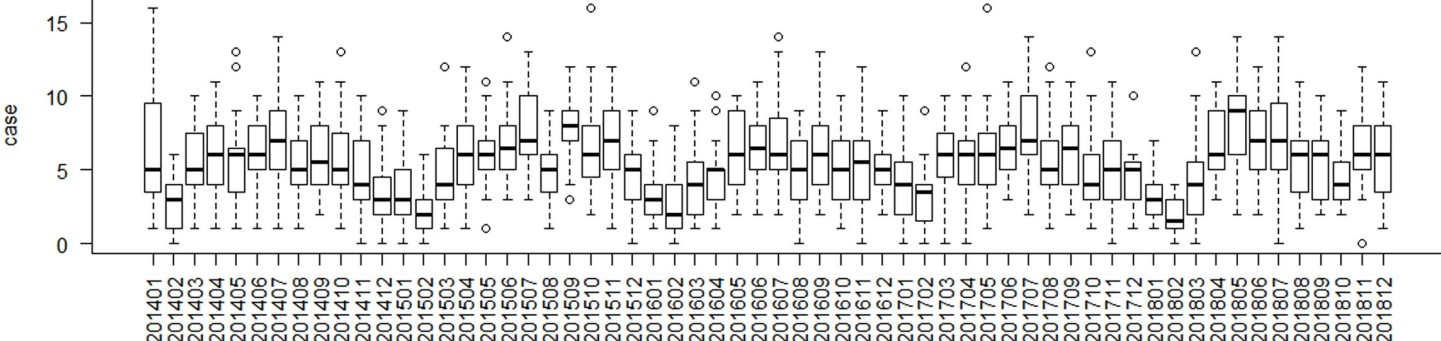

**Fig 2. The boxplot of mumps cases among children in Guangzhou, 2014–2018.**

## Results

From January 1st, 2014 to December 31st, 2018, a total of 9842 mumps cases among persons younger than 18 years were reported in Guangzhou. During 2014 to 2018, the breakdown of mumps cases was 1962, 2077, 1838, 1977, and 1988, respectively. The gender ratio was 1.83 (6362 males to 3480 females). Among the groups aged 0–3, 4–6, 7–14, and 15–17 years, there were 2197 (22.32%), 3773 (38.34%), 3672 (37.31%), and 200 (2.03%) cases, respectively. The peak of mumps occurrence in the study period occurred from May to July, with a total of 3087 cases (31.37%) (Fig 2).

A description of meteorological variables and mumps cases is given as follows (Fig 3). The mean daily temperature was 22.24˚C, and the means of diurnal temperature, aggregate rainfall, atmospheric pressure, wind speed, relative humidity, and hours of sunshine were 8.20˚C, 6.04 mm, 1005.41 hPa, 2.16 m/s, 79.19%, and 4.42 hours, respectively. The monthly distribution is shown in the monthly boxplot of meteorological factors (Fig 4). The results of the Mann–Kendall and Pettitt tests are shown in Table 1. The variables of DTR, rain, pressure, wind, relative humidity were statistically significant. Wind and relative humidity had a rising tendency, whereas the DTR, rain and pressure had a decrease tendency. No tendency was observed among the variables of cases, temperature and sunshine hours.

Fig 5 shows the interaction network between meteorological factors and mumps cases among children, the strongest interaction was observed between temperature and mumps cases. Temperature, sunshine hour, relative correlation and aggregate rainfall had a positive interaction with mumps cases, whereas atmosphere pressure, wind speed and DTR had a negative interaction with mumps cases. Spearman's correlation analyses revealed that the incidence of mumps was positively correlated with mean temperature, relative humidity, aggregate rainfall, and hours of sunshine. On the contrary, wind speed and atmospheric pressure were negatively correlated with the incidence of mumps. Mean temperature was strongly correlated with atmospheric pressure ($r = -0.82$, $P<0.01$).

Fig 6 illustrates the correlation between mumps cases and meteorological variables with different lag periods in Guangzhou. Specific features were observed in various variables with a nonlinear curve. The highest relative risk (RR) was 1.005 (95% confidence interval [CI]: 1.001–

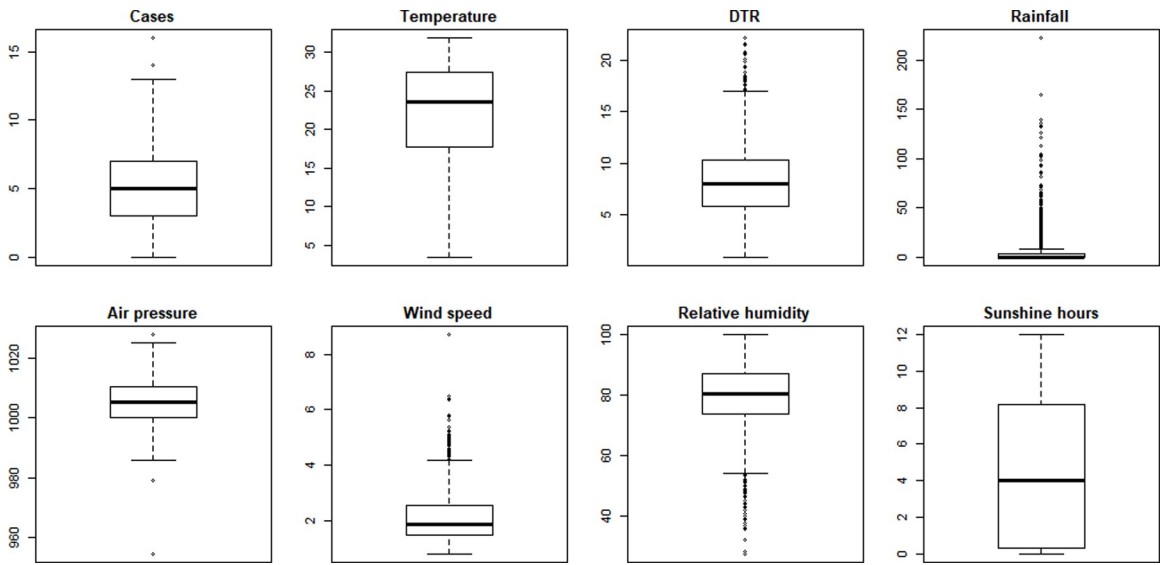

**Fig 3. The boxplot of meteorological factors and mumps cases among children in Guangzhou, 2014–2018.**

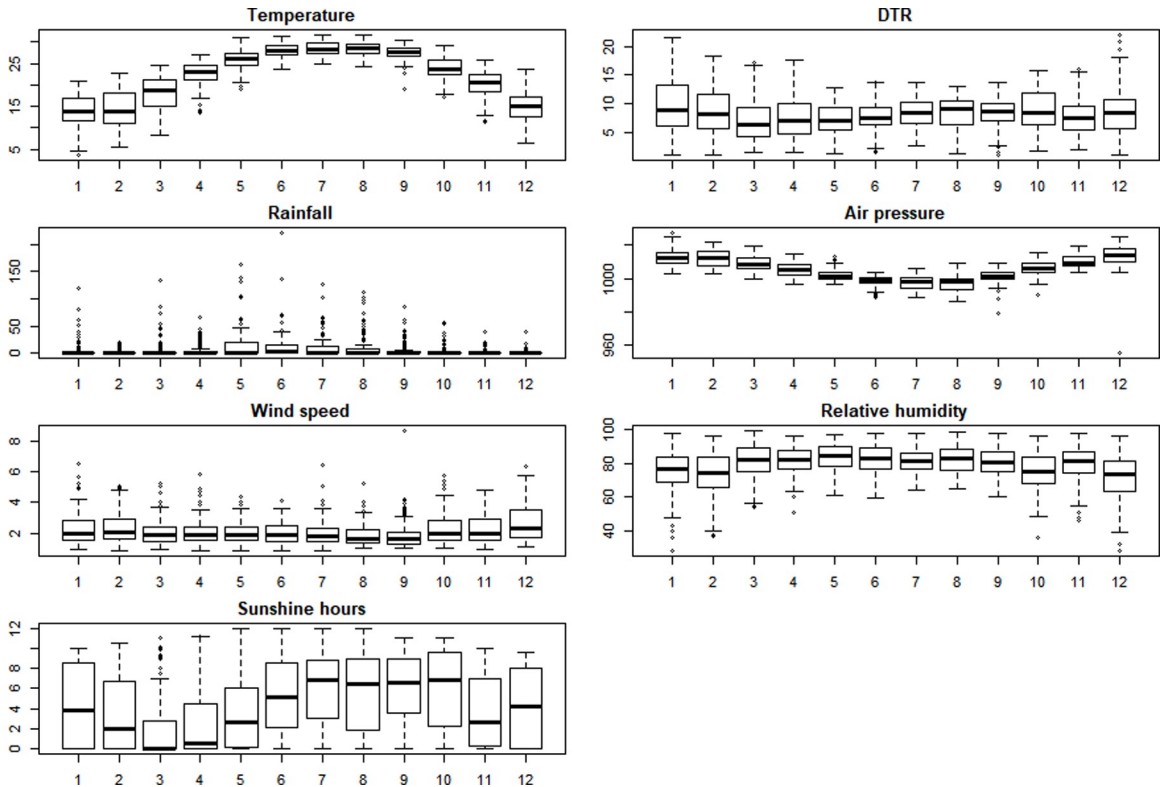

**Fig 4. The monthly boxplot of meteorological factors in Guangzhou, 2014–2018.**

1.010) at lag day 21, when the daily mean temperature was 25.0°C. When atmospheric pressure was 1027 hPa at lag day 0, the lowest RR was 0.775 (95% CI: 0.655–0.916). When relative humidity was 72.0%, the highest RR was 1.014 (95% CI: 1.003–1.026) at lag day 12. When wind velocity was 8.70 m/s, the highest RR was 1.269 (95% CI: 1.033–1.560) at lag day 11.

For a better explanation, the one-dimensional curves were plotted about the overall effects of meteorological factors over the corresponding lag days (Fig 7). In a word, mean temperature, atmospheric pressure, and relative humidity were associated with mumps incidence, unlike wind speed. The RR of mean temperature significantly increased from 10°C to 23.6°C, whereas that of wind speed significantly declined from 1.9 to 4.5 m/s. Atmospheric pressure

**Table 1. The results of the Man-Kendall and Pettitt tests for the meteorological variables and mumps cases among children in Guangzhou, 2014–2018.**

| variables | P | S | change-point number | change-point | P |
|---|---|---|---|---|---|
| case | 0.3 | + | 1542 | 2018/3/22 | <**0.01** |
| temperature | 0.19 | + | 129 | 2014/5/9 | <**0.01** |
| DTR | <**0.01** | - | 414 | 2015/2/18 | <**0.01** |
| rain | <**0.01** | - | 986 | 2016/9/12 | <**0.01** |
| pressure | <**0.01** | - | 482 | 2015/4/27 | <**0.01** |
| wind | <**0.01** | + | 371 | 2015/1/6 | <**0.01** |
| relative humidity | <**0.01** | + | 794 | 2016/3/4 | <**0.01** |
| Sunshine hours | 0.88 | - | 674 | 2015/11/5 | **0.02** |

\# S: '+' means a rise tendency, '-'means a decrease tendency.

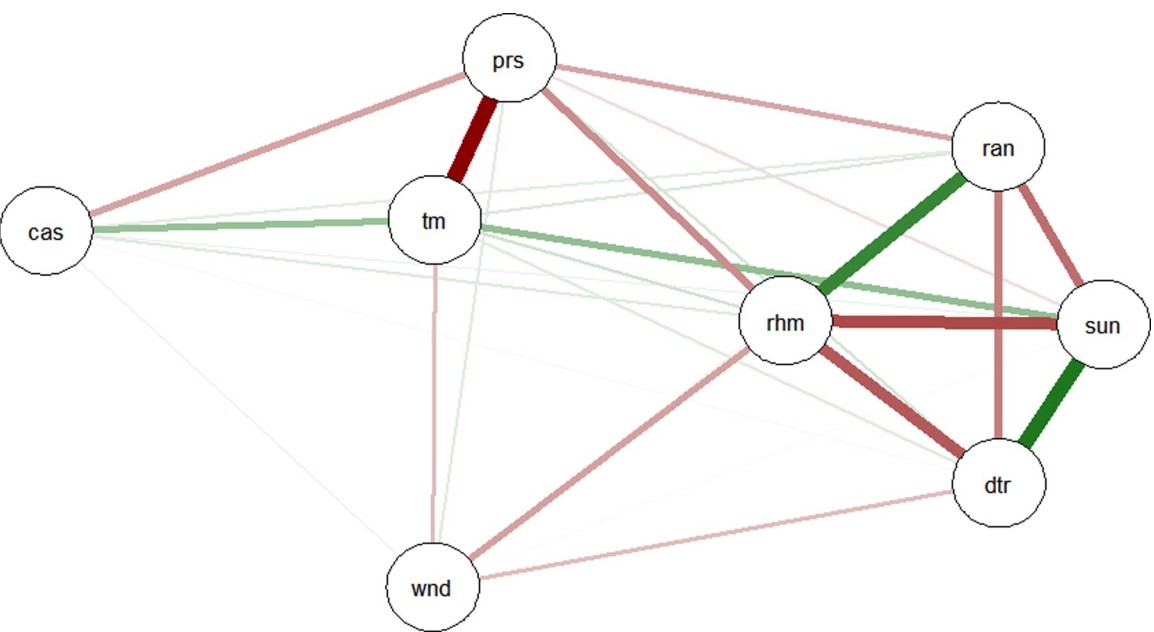

**Fig 5. The interaction network between meteorological factors and mumps cases among children in Guangzhou, 2014–2018.** *cas represents cases; tm represents temperature; prs represents atmospheric pressure; wnd represents wind; dtr represents diurnal temperature range; ran represents rain; rhm represents relative humidity; sun represents sunshine hours. The green line means positive relationship, whereas the red one means negative relationship. The width of the lines dedicates the degree of *r*. The thicker is the line, the greater is the interaction.

and relative humidity both showed a reverse U-shaped curve relationship with mumps. However, the RRs just had a significantly increasing trend before the atmosphere pressure as 992

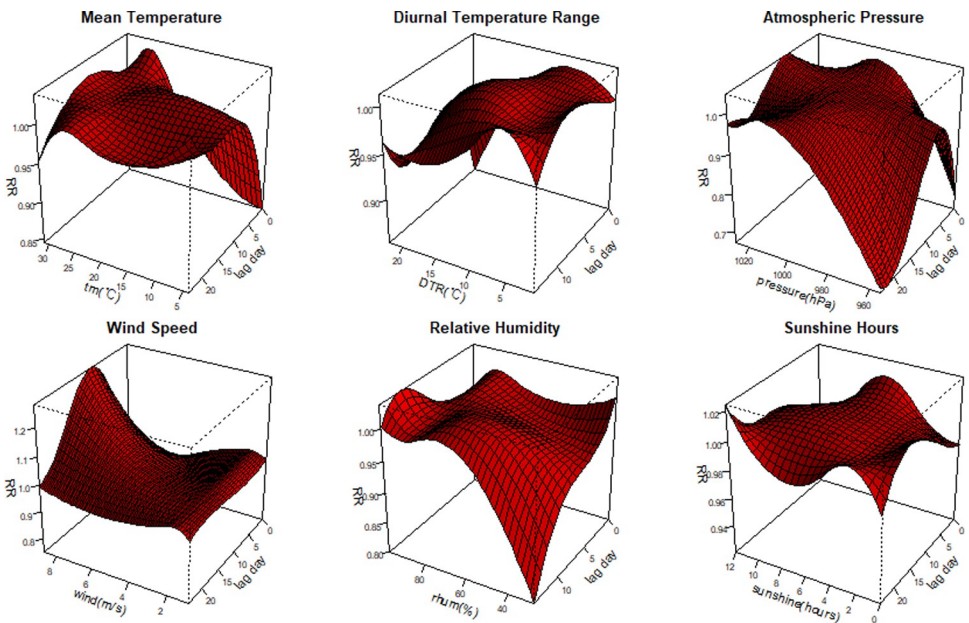

**Fig 6. Plots of the relative risk of climatic factors on the incidence of mumps in children, including mean temperature, diurnal temperature range, atmospheric pressure, wind speed, relative humidity and sunshine hours.**

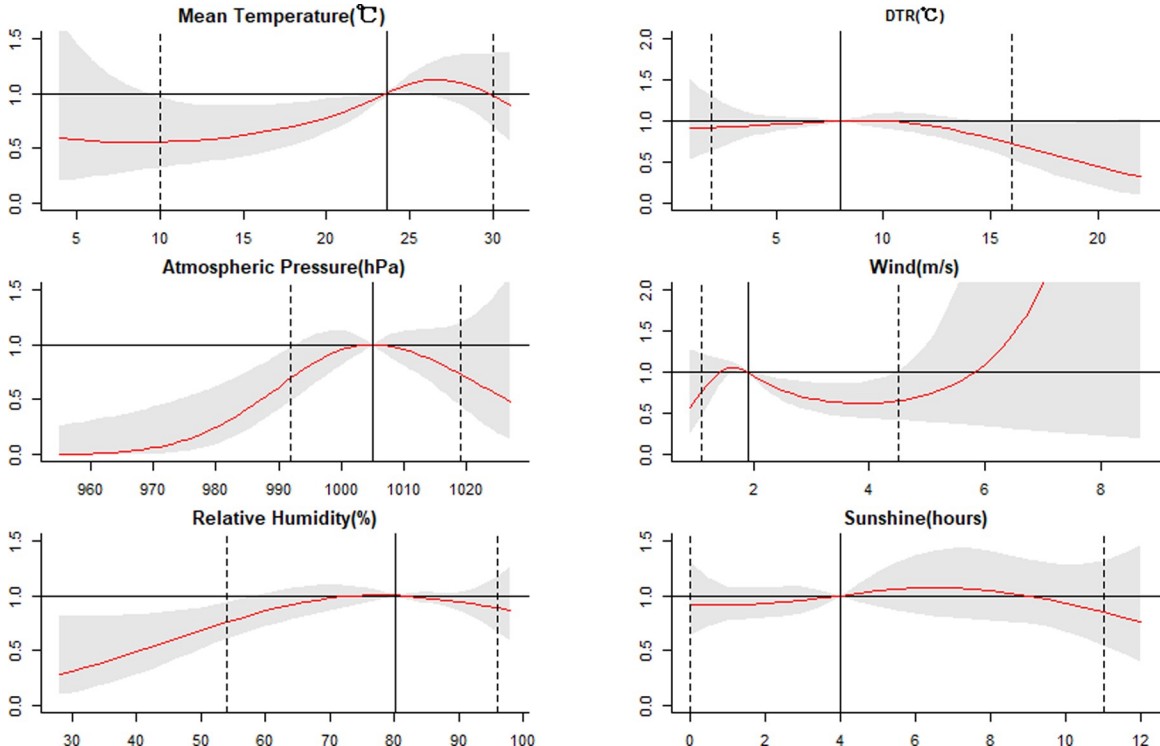

**Fig 7. The estimated overall effects of mean temperature, diurnal temperature range, atmospheric pressure, wind speed, relative humidity, and sunshine hours along corresponding lag days.** The Y lab represents the value of relative risk, the X lab represents the value of relevant variables. The red lines represent mean relative risks and the grey regions represent 95% CIs.The black vertical line represents the medians of the climatic factors, and the dotted lines represent the 2.5 percentile and the 97.5 percentile for the meteorological factors, respectively.

hPa and the relative humidity as 60%, respectively. There were no significant relationships between other meteorological variables and the incidence of mumps in our study.

The cumulative extreme effects of meteorological factors on mumps along the corresponding lag days are shown in **Fig 8**. As shown in **Table 2**, the extreme low effect of temperature was significant and the lowest RR was 0.88 (95% CI: 0.83–0.93) at lag day 0, whereas the extreme high effect of temperature was not significant. On the contrary, the extreme high effects of atmospheric pressure and wind speed were statistically significant, whereas the extreme low effects were not. For atmospheric pressure, the lowest RR was 0.88 (95% CI: 0.82–0.94) at lag day 0. For wind speed, the lowest RR was 0.94 (95% CI: 0.89–0.98) at lag day 0. Regarding relative humidity, the RR of the extreme high effect was 0.96 (95% CI: 0.94–0.99) at lag day 2, and the RR of the extreme low effect was 0.96 (95% CI: 0.93–0.99) at lag day 2. For the extreme high and low effects of sunshine hours, the RRs were 1.03 (95% CI: 1.02–1.04) at lag day 2 and 1.02 (95% CI: 1.01–1.03) at lag day 17, respectively.

In the analyses by subpopulation, the extreme low effects of temperature, atmospheric pressure, and relative humidity, and the extreme high effect of wind speed were statistically significant in the female population, but no significant extreme effect was observed in the male population. The extreme low effect of temperature was observed in the 4–6 year old group (RR = 0.25, 95% CI: 0.11–0.60) and 15–17 year old group (RR = 0.01, 95% CI: 0.00–0.57). The extreme high effect of atmospheric pressure was observed in the 4–6 year age group (RR = 0.46, 95% CI: 0.21–0.98) (**Table 3**).

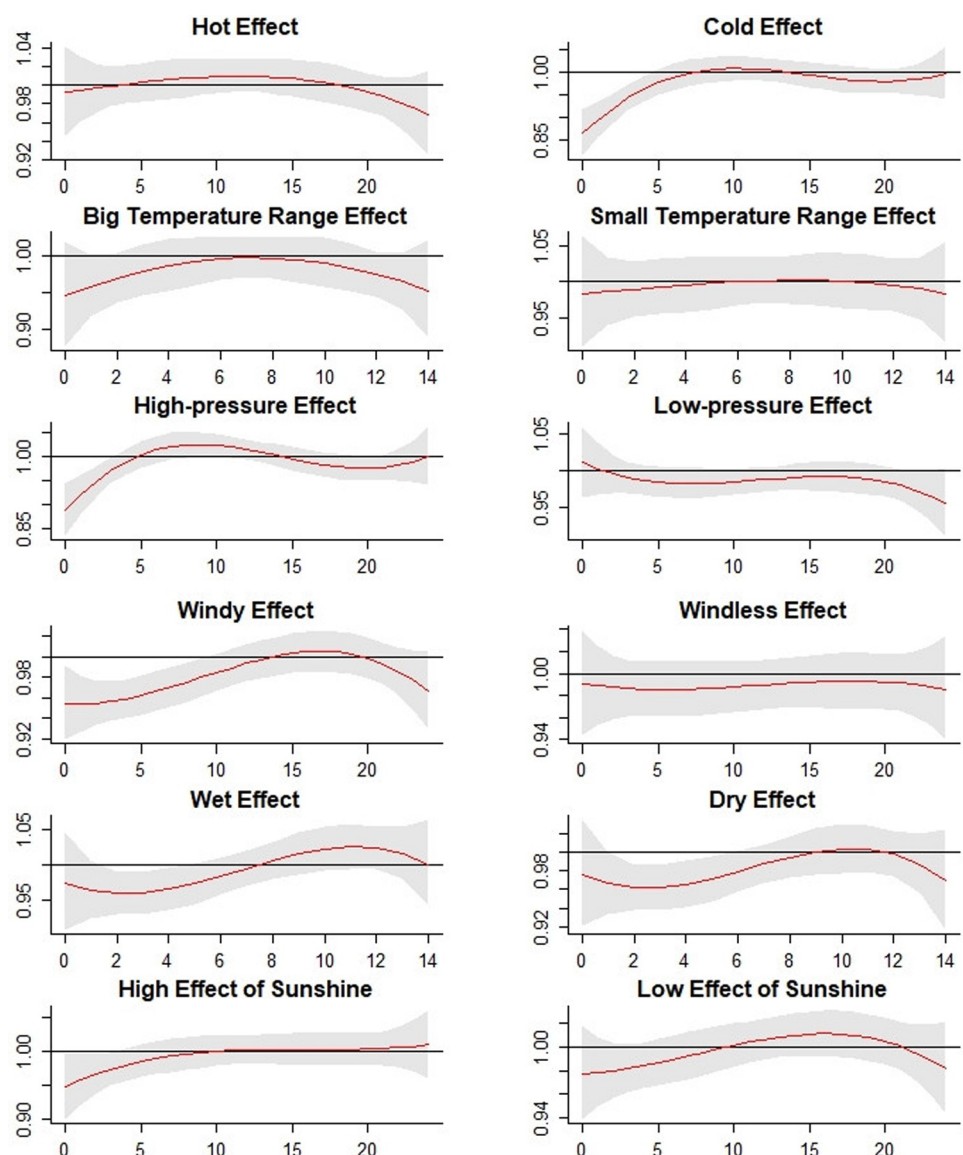

**Fig 8. The extreme effects of mean temperature, diurnal temperature range, atmospheric pressure, wind speed, relative humidity, and sunshine hours with extreme high effects (97.5%) and extreme low effects (2.5%).** The Y lab represents the value of relative risk, the X lab represents the value of lag day. The red lines represent mean relative risks and grey regions represent 95%CIs.

**Table 2. The extreme effects of meteorological variables on the incidence of mumps among children along the lag days in Guangzhou, 2014–2018.**

| Meteorological variables | Lag day | High effect (97.5%) | | Lag day | Low effect (2.5%) | |
|---|---|---|---|---|---|---|
| | | RR | 95%CI | | RR | 95%CI |
| Mean temperature | / | / | / | 0 | 0.88 | 0.83–0.93 |
| Atmospheric pressure | 0 | 0.88 | 0.82–0.94 | / | / | / |
| Wind speed | 0 | 0.94 | 0.89–0.98 | / | / | / |
| Relatively humidity | 2 | 0.96 | 0.94–0.99 | 2 | 0.96 | 0.93–0.99 |

**Table 3. The cumulative extreme effects of meteorological variables on mumps cases of children by sex and age.**

| Variables | Mean temperature | | | | Wind speed | | | | Relative humidity | | | | Atmospheric pressure | | | |
|---|---|---|---|---|---|---|---|---|---|---|---|---|---|---|---|---|
| | low effect | | high effect | | low effect | | high effect | | low effect | | high effect | | low effect | | high effect | |
| | RR | 95%CI | RR | 95%CI | RR | 95%CI | RR | 95%CI | RR | 95%CI | RR | 95%CI | RR | 95%CI | RR | 95%CI |
| total | **0.56** | **0.32–0.99** | 0.99 | 0.70–1.40 | 0.77 | 0.47–1.28 | 0.64 | 0.40–1.01 | **0.76** | **0.61–0.94** | 0.89 | 0.67–1.17 | **0.70** | **0.49–0.99** | 0.75 | 0.45–1.24 |
| male | 0.88 | 0.44–1.73 | 0.91 | 0.59–1.39 | 0.76 | 0.41–1.41 | 0.88 | 0.50–1.56 | 0.95 | 0.73–1.23 | 1.06 | 0.75–1.49 | 0.87 | 0.56–1.34 | 0.81 | 0.43–1.50 |
| female | **0.23** | **0.10–0.62** | 1.12 | 0.66–2.02 | 0.77 | 0.74–1.71 | **0.34** | **0.16–0.72** | **0.50** | **0.35–0.72** | 0.64 | 0.41–1.00 | **0.47** | **0.26–0.83** | 0.64 | 0.28–1.45 |
| <4years | 1.83 | 0.60–5.65 | 1.40 | 0.69–2.83 | 0.99 | 0.36–2.71 | 0.74 | 0.29–1.87 | 0.73 | 0.48–1.11 | 0.70 | 0.39–1.24 | 0.54 | 0.26–1.09 | 0.97 | 0.37–2.55 |
| 4–6 years | **0.25** | **0.11–0.60** | 1.01 | 0.58–1.75 | 0.68 | 0.31–1.49 | 0.51 | 0.25–1.05 | 0.77 | 0.56–1.07 | 1.28 | 0.83–1.96 | 0.81 | 0.46–1.42 | **0.46** | **0.21–0.98** |
| 7–13 years | 0.82 | 0.32–2.10 | 0.83 | 0.48–1.44 | 0.79 | 0.35–1.80 | 0.81 | 0.37–1.76 | 0.76 | 0.52–1.11 | 0.72 | 0.45–1.14 | 0.73 | 0.41–1.30 | 1.25 | 0.51–3.05 |
| 14–17 years | **0.01** | **0.00–0.57** | 0.18 | 0.01–2.74 | 0.35 | 0.01–16.28 | 0.05 | 0.00–2.18 | 0.48 | 0.07–3.29 | 0.38 | 0.05–3.09 | 0.20 | 0.01–2.66 | 0.08 | 0.00–3.08 |

Sensitivity analyses indicated similar results after the changes in the df for long-term trends and seasonality. Altering the dfs for temperature, atmospheric pressure, wind speed, and relative humidity, the original models showed better robustness compared to their alternatives.

## Discussion

In this study, the incidence of mumps among children in Guangzhou fluctuated seasonally with the epidemic peak from May to July. The trough of mumps incidence among children occurred in February, coinciding with the winter holiday in school. However, our result differed from that of a study conducted in Ireland, which detected a relatively low mumps incidence in summer and a peak in autumn [7], and a study conducted in the Netherlands with two seasonality peaks in spring and autumn [26]. In China, the epidemic pattern in Guanghzou also differed from that of Beijing city, Zibo city in Shandong province, and Chongqing city, which showed two epidemic peaks from April to July and October to January, with an emphasis on the former [17, 27, 28]. It indicated that the effect of meteorological factors on the occurrence of mumps may be distinct in various regions with particular climatic features.

Climatic variables such as temperature, relative humidity, and wind speed have been proven to play a key role in the occurrence of certain infectious diseases [29–31]. Recently, more and more attention has been drawn to the effects of meteorological factors on mumps incidence. Some studies have been conducted to explore the relationships between meteorological factors and mumps cases; however, most of the studies just detected linear relationships without lag effects. Recently, DLNMs have been applied widely to evaluate these nonlinear correlations with lag effects [10, 21, 23]. However, these relationships among children still need to be explored.

To the best of our knowledge, this is the first study to apply DLNMs to detect the effects of meteorological variables on mumps incidence among children in a subtropical city in China. Nonlinear relationships were observed in some meteorological variables in our study. When the mean temperature ranged from 10–23.6˚C, the RR showed an increased trend. Our finding was not fully consistent with previous studies. A study conducted in Mexico showed that the mumps incidence declined markedly during the summer [32]. However, a study conducted in Taiwan [33] reported that the incidence of mumps started to increase at a temperature of 20˚C, and started to decline after 25˚C, whereas a study conducted in Jining, China [18] and a study focused on children in Japan [24] both reported linearly positive associations between mumps incidence and temperature. A previous study conducted in Guangzhou [22] also revealed a positive correlation between mean temperature and the incidence of mumps in the overall population. However, another study conducted in Jining showed a U-shaped relationship curve between mean temperature and mumps incidence [34]. The discrepancy may be due to the differences in methodology, geographic location, and climatic characteristics. The

incidence of mumps depended on the contact rate, which might be influenced by social behavior [35]. A previous epidemiologic study conducted in Canada showed that adolescents usually engage in more physical activity in the warmer months than in the colder months [36]. Children are more likely to come into contact with each other in warmer weather, which might result in an increase in the incidence of mumps. When the temperature continued to increase, outdoor activities among children might decrease, reducing the risk of mumps infection; however, the extreme high effect was not statistically significant. For the same reason, the risk of mumps infection will be reduced in extreme cold conditions; however, the accumulated extreme cold effect was only observed significantly among females. These might explain our finding among children in Guangzhou and shows that preventative measures and health education should be strengthened in the temperature range mentioned above in our study.

The results of some studies were inconsistent with ours. For example, relative humidity was found to have a negative impact on mumps incidence with a threshold of 54% in Jining, China [18]. However, another study conducted in Jining showed that the effect of relative humidity on mumps increased slightly and then decreased quickly, with a threshold of 64% [34]. Besides, a study conducted in Taiwan [33] found no correlation between relative humidity and mumps incidence. Our study showed a U-shaped relationship between relative humidity and mumps incidence among children in Guangzhou, with a statistically significant increase before a relative humidity of 60%. Meanwhile, nonlinear extreme high and low effects of relative humidity on mumps incidence were both found among children. Studies on the mechanism underlying the effect of relative humidity on mumps incidence are rare and should be carried out in the future. One possible mechanism might be as follows. Droplet spread is one of the modes of transmission of the mumps virus [1]. When the relative humidity is low, water in the droplets evaporates quickly. The longer the droplet with mumps viruses remains airborne, the easier it is the mumps infect the children [37]. As the relative humidity increases, the effects become insignificant. However, an extremely high effect of relative humidity was observed with an RR of 0.96 (95% CI: 0.93–0.99) at lag day 2, which may be explained by the fact that high relative humidity often coincides with heavy rain, which will make children more likely to stay at home [38] and reduce the contact rate. More studies should be carried out to identify biological evidence.

Wind could affect the suspension time of viruses and their diffusion distance [39, 40]. Our study revealed that the RR decreased as the wind speed surpassed 1.9 m/s till 4.9 m/s. Moreover, the high effect of wind speed yielded an RR of 0.94 (95% CI: 0.89–0.98). Our study result was similar to that of a study conducted in Shenzhen, which depicted a reverse V-shaped relationship between wind velocity and mumps incidence, with the peak at 2 m/s [21], and a study conducted in Fujian province [23], which showed a negative relationship between wind speed and mumps incidence. In view of the resuspension phenomenon, the concentration of local viruses may be diluted by higher wind velocity [39, 41], which could possibly explain our finding. In the subpopulation analysis, an accumulated high effect of wind speed was observed among females. This may be explained by the reduction in outdoor activity among females in the extreme high wind velocity weather. Based on our findings, regular ventilation might be helpful in protecting children against infection with mumps.

In our study, extreme low effects of mean temperature, atmospheric pressure, and relative humidity, and an extreme high effect of wind speed were found among females with RR values less than 1, whereas no extreme effect was detected among males. This may be due to factors such as physiological gender differences and the discrepancy in behavior patterns [42]. For example, in extreme weather conditions, males are more active and will be more likely to play with friends, which will increase the contact rate with children infected with mumps. This shows that children should cut down on going to crowded places such as playgrounds in the active period of mumps from May to July in Guangzhou.

Our results showed a positive correlation before 1000.05 hPa, after which point the correlation was not statistically significant. However, previous studies showed different results. For instance, a study conducted in 10 cities in Guangxi province in China [20] reported that no significant relationship was found between atmospheric pressure and the incidence of mumps, whereas a study conducted in Beijing [17] showed that atmospheric pressure was negatively associated with mumps incidence. Unfortunately, limited studies are available regarding the underlying mechanism. This may be explained as follows. As the atmospheric pressure increases in the beginning, the cold air would enter and promote the spread of the virus, leading to high virus abundance in the atmosphere. After exceeding a certain value of atmospheric pressure, the effect mentioned above is offset by high wind speed. However, further studies should be conducted to explore the foundational mechanism in future.

Several limitations of our study should be acknowledged. At first, the data were only obtained in Guangzhou, southern China; hence, it cannot stand for other cities or countries. Secondly, we used meteorological data instead of the exposure conditions of individuals; this may lead to bias in our results. Further, we could not exclude the possibility that some potential confounding factors existed in our research, including host susceptibility, preventive measures, geographic factors, and population density. Moreover, data regarding human serum antibodies against mumps and information regarding mumps vaccination were unavailable in our study. Finally, we did not assess factors regarding air pollution which have been proven to affect the incidence of mumps [43]. Therefore, more studies should be undertaken to take these variables into consideration.

## Conclusions

In general, compared to previous studies that focused on the effect of meteorological factors on mumps incidence, for the first time, we identified a significant nonlinear association between certain climatic variables and mumps incidence among children in Guangzhou with lag effects. At a certain corresponding range, mean temperature, atmospheric pressure, and relative humidity were positively correlated with the incidence of mumps among children, opposite to that of wind speed. Moreover, extreme effects of the variables mentioned above among mumps cases were detected in a subgroup analysis according to gender and age. These findings provide the underlying information to better understand the epidemic tendency of mumps. Based on our results, some measures could be implemented to minimize the occurrence of mumps. For example, preventative measures and health education should be strengthened in the temperature range of 10–23.6˚C, and proper ventilation might be helpful in protecting children against infection with mumps. We also implied that children should cut down on going to crowded places such as playgrounds in the active period of mumps from May to July in Guangzhou. Our findings in this study will help improve the early warning system and reinforce the prevention and intervention of risk factors for mumps among children in Guangzhou.

## Supporting information

**S1 Data. The data and the R program of our study were uploaded as the supporting information file, which also included the data of subpopulation analyses.**
(ZIP)

## Acknowledgments

We thank the Guangzhou Meteorological Bureau for offering climatic data for our study. We would like to thank Editage (www.editage.cn) for English language editing.

## Author Contributions

**Data curation:** Jianyun Lu.

**Methodology:** Jianyun Lu, Xiaowei Ma.

**Software:** Xiaowei Ma.

**Supervision:** Zhicong Yang, Zhoubin Zhang.

**Visualization:** Mengmeng Ma.

**Writing – original draft:** Jianyun Lu.

**Writing – review & editing:** Zhicong Yang, Zhoubin Zhang.

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
