## [Decision Letter · Decision Letter 0]

9 Jan 2020

PONE-D-19-31388

The roles of meteorological factors on mumps incidence among children in Guangzhou, Southern China

PLOS ONE

Dear Dr. Zhang,

Thank you for submitting your manuscript to PLOS ONE. After careful consideration, we feel that it has merit but does not fully meet PLOS ONE’s publication criteria as it currently stands. Therefore, we invite you to submit a revised version of the manuscript that addresses the points raised during the review process.

Dear authors, your manuscript has been reviewed by two peer reviewers. Both requested Major Revisions to consider the manuscript to be published in Plos One.

I agree with all the notes made by the reviewers.

You must also review English. I suggest specialized companies like Editage and American Journal Experts.

In addition, I submit some mandatory corrections for the manuscript to be submitted to a new review round by the reviewers.

Therefore, I invite the authors to respond point-by-point to each comment made by the reviewers and myself (see below).

We would appreciate receiving your revised manuscript by Feb 10 2020 11:59PM. To enhance the reproducibility of your results, we recommend that if applicable you deposit your laboratory protocols in protocols.io, where a protocol can be assigned its own identifier (DOI) such that it can be cited independently in the future. For instructions see: http://journals.plos.org/plosone/s/submission-guidelines#loc-laboratory-protocols

We look forward to receiving your revised manuscript.

Kind regards,

Paulo Eduardo Teodoro, Prof. Dr.

Academic Editor

PLOS ONE

Additional Editor Comments:

Dear authors, your manuscript has been reviewed by two peer reviewers. Both requested Major Revisions to consider the manuscript to be published in Plos One.

I agree with all the notes made by the reviewers.

You must also review English. I suggest specialized companies like Editage and American Journal Experts.

In addition, I submit some mandatory corrections for the manuscript to be submitted to a new review round by the reviewers.

Therefore, I invite the authors to respond point-by-point to each comment made by the reviewers and myself (see below).

- Abstract should be written according to Plos One models;

- It is necessary to include at the beginning of the Abstract and at the end of the Introduction what are the hypotheses of this research. This can be done in the form of a question;

- Keywords should be better chosen. They should not contain words that are in the title of the manuscript;

- There are several manuscripts addressing the theme "meteorological factors on mumps". In a quick search (https://www.sciencedirect.com/search/advanced?qs=meteorological%20factors%20mumps) I found at least seven similar manuscripts. Therefore, the authors should make clear in the Introduction what is the difference in the work done for papers that are already published. This needs to be improved in the Introduction;

- In the Material and Methods is necessary a map of the study area location;

- Although the authors employed robust statistical analyzes, further testing is still needed to improve the discussion of results. As the authors worked with an 18-year time series, it is necessary to check for changes in this time series with the nonparametric Man-Kendall and Pettitt tests. These tests are easily performed in the software R and may show if there was a tendency to increase or decrease in any meteorological variable and in the cases;

- Table 1 is important, but I would like to see a boxplot for these variables. Use the ggplot2 package;

- It is necessary to inform which test applied the Sperman correlations. In addition, I invite the authors to graphically express the results in Table 2 through the correlation network. It is a didactic way of representing this matrix. Can be done with the "qgraph" package in R;

- It is necessary to inform in the legend of Figures 2, 3 and 4 if the adjusted models were significant and which test was used for this;

- Discussion is very poor and needs to be improved. Authors should include findings from recent manuscripts I mentioned;

- In the conclusions, it is necessary to inform readers about the scientific advances that this manuscript has obtained in relation to the works already published. What are the proposals to minimize the occurrence of mumps in children by knowing the behavior of meteorological variables?

- The authors used only 35 references and most of them are old. In addition to being very little, recent references need to be included to demonstrate to readers that while there are many published manuscripts, the topic still needs to be studied.

Journal Requirements:

2. In the ethics statement in the manuscript and in the online submission form, please provide additional information about the patient records used in your retrospective study.

Specifically, please ensure that you have discussed whether all data were fully anonymized before you accessed them and/or whether the IRB or ethics committee waived the requirement for informed consent.

If patients provided informed written consent to have data from their medical records used in research, please include this information.

Reviewers' comments:

Reviewer's Responses to Questions

**Comments to the Author**

1. Is the manuscript technically sound, and do the data support the conclusions?

Reviewer #1: Partly

Reviewer #2: Yes

2. Has the statistical analysis been performed appropriately and rigorously? 

Reviewer #1: Yes

Reviewer #2: Yes

3. Have the authors made all data underlying the findings in their manuscript fully available?

Reviewer #1: Yes

Reviewer #2: No

4. Is the manuscript presented in an intelligible fashion and written in standard English?

Reviewer #1: No

Reviewer #2: Yes

5. Review Comments to the Author

Reviewer #1: Dear authors,

The subject of the manuscript is interesting and topical, especially for the return of diseases previously eradicated or controlled in the last century. However, I found several weaknesses in the manuscript, for example, presentation of the results needs to improve, the discussion of the results does not cover other continents, only being restricted to Asia and Europe. As well as several gross errors throughout the manuscript, implying that the manuscript was not revised correctly. It is necessary to improve the quality of the figures to make them self explanatory. It is necessary to improve the conclusions of the manuscript.

Reviewer #2: Dear Authors,

The manuscript makes an interesting based on interaction between mumps and meteorological variables. It is believed that more information on the subject can still be extracted, even in face of the limitations described by authors.

On this aspect, some points need to be improved throughout the text, I believe they should include the references of R software, as well as dlnm package, both described on manuscript.

The introduction and materials and methods are well described.

Regarding the results, I have some suggestions for improvement:

1) Replacement of Figure 1 with a boxplot of mumps cases on an annual and monthly basis.

2) The other variables would be displayed in Figure 2, with monthly boxplots, making the visualization of the results much more understandable in some passages of the text, exploring the maximum and minimum.

3) The old Figure 2 (which will be for Figure 3) could include the intensities, not only leaving the color red, this would show the interaction between them.

4) Improve the image resolution of all images, as they are of low quality.

5) Paragraph preceding Figure 3, correct the pHa for HPa, on page 11.

Well, from these improvements and implementations, the article will be able to be approved.

6. PLOS authors have the option to publish the peer review history of their article (what does this mean?). If published, this will include your full peer review and any attached files.

Reviewer #1: No

Reviewer #2: No

---

## [Author Response · Author response to Decision Letter 0]

20 Feb 2020

Dear Editor,

We acknowledge with thanks the receipt of the comments of the reviewer. These comments/requirements were highly beneficial in the modification of the manuscript. The manuscript was revised according to all these comments. All modifications and/or corrections are highlighted by using the track changes mode. We would like to thank Editage (www.editage.cn) for English language editing. A point-to-point response to the reviewers’ comments was attached. We appreciate you kindly offer the opportunity to transfer our manuscript.

Additional Editor Comments

Question No1: Abstract should be written according to Plos One models;

Response to the question No.1: We re-wrote the Abstract according to Plos One models.

Question No.2：It is necessary to include at the beginning of the Abstract and at the end of the Introduction what are the hypotheses of this research. This can be done in the form of a question;

Response to the question No.2: We added the hypotheses of this research in the article ,L81-86“However, the nonlinear and lag effects of climatic factors on the transmission among children, who are the main population affected by mumps, are still unclear at present. A study conducted in Japan explored the linear relationship between meteorological factors and mumps incidence among children via negative binomial regression analysis (24). What are the roles of meteorological factors on mumps incidence among children? There is an urgent need to explore these nonlinear relationships using lag effects..”

Question No.3： Keywords should be better chosen. They should not contain words that are in the title of the manuscript;

Response to the question No.3: Thanks for your question. As your suggestion, the Keywords were revised as following:“distributed lag nonlinear models, climatic factors, effect”.

Question No.4：There are several manuscripts addressing the theme "meteorological factors on mumps". In a quick search (https://www.sciencedirect.com/search/advanced?qs=meteorological%20factors%20mumps) I found at least seven similar manuscripts. Therefore, the authors should make clear in the Introduction what is the difference in the work done for papers that are already published. This needs to be improved in the Introduction;

Response to the question No.4: Thanks for your question. As your suggestion, we described the difference of the similar researches on mumps, and the section were revised as L68-86.

Question No.5： In the Material and Methods is necessary a map of the study area location;

Response to the question No.5: The map of the study area location was added in the Material and Methods.

Question No.6： Although the authors employed robust statistical analyzes, further testing is still needed to improve the discussion of results. As the authors worked with an 18-year time series, it is necessary to check for changes in this time series with the nonparametric Man-Kendall and Pettitt tests. These tests are easily performed in the software R and may show if there was a tendency to increase or decrease in any meteorological variable and in the cases;

Response to the question No.6: We use the nonparametric Man-Kendall and Pettitt tests to check for changes in our time series data. And the result was shown as Table 1

Question No.7：Table 1 is important, but I would like to see a boxplot for these variables. Use the ggplot2 package; 

Response to the question No.7：Thanks for your suggestion. We used the boxplot (Fig 3.) to describe these meteorological variables instead of the original table.

Question No.8：It is necessary to inform which test applied the Sperman correlations. In addition, I invite the authors to graphically express the results in Table 2 through the correlation network. It is a didactic way of representing this matrix. Can be done with the "qgraph" package in R;

Response to the question No.8: Thanks for your suggestion. We applied the correlation network(Fig 5.)to graphically express the results in the original Table 2.

Question No.9：It is necessary to inform in the legend of Figures 2, 3 and 4 if the adjusted models were significant and which test was used for this. 

Response to the question No.9: Based on the AICs, we selected the smallest AIC to different meteorological variables to structure the best models. We showed the RR and 95% confidence interval (95%CI) of the original figure 2，3 and 4. Meantime, we could see the RR and 95% CI in the original figure 3 and 4 (The red lines represent mean relative risks and the grey regions represent 95%CIs.) If the 95% CI include “1” as the RR value, there is not significant difference. Otherwise, it means that it’s significantly different. It equals to the P value <0.05. 

Question No.10：Discussion is very poor and needs to be improved. Authors should include findings from recent manuscripts I mentioned; 

Response to the question No.10: Thanks for your advice. We got some recent manuscripts the reviewer mentioned and other new article into reference added to 43 articles, and revised the discussion. 

Question No.11：In the conclusions, it is necessary to inform readers about the scientific advances that this manuscript has obtained in relation to the works already published. What are the proposals to minimize the occurrence of mumps in children by knowing the behavior of meteorological variables? 

Response to the question No.11:

In the conclusions, scientific advance was informed as L358-361 “In general, compared to previous studies that focused on the effect of meteorological factors on mumps incidence, for the first time, we identified a significant nonlinear association between certain climatic variables and mumps incidence among children in Guangzhou with lag effects.” And L363-365“Moreover, extreme effects of the variables mentioned above among mumps cases were detected in a subgroup analysis according to gender and age.” Due to the results of our study, some proposals were recommended to minimize the occurrence of mumps among children. L367-371 “For example, preventative measures and health education should be strengthened in the temperature range of 10–23.6°C, and proper ventilation might be helpful in protecting children against infection with mumps. We also implied that children should cut down on going to crowded places such as playgrounds in the active period of mumps from May to July in Guangzhou.”

Question No.12：The authors used only 35 references and most of them are old. In addition to being very little, recent references need to be included to demonstrate to readers that while there are many published manuscripts, the topic still needs to be studied.

Response to the question No.12: Thanks for your kindly reminder, We reviewed the recent references and revised the introduction, and demonstrated the scientific advance and the difference between our study and previous researches. Firstly our study use the DLNMs to explore the nonlinear relationships with lag effects between meteorological variables and mumps cases. Secondly we firstly used the DLNMs to analyze the meteorological effects on mumps cases among children. Finally, we analyze the effect by subpopulation by sex and age among children. This detail could be seen in the “introduction and discussion”. 

“ 

Reviewer #1: 

The subject of the manuscript is interesting and topical, especially for the return of diseases previously eradicated or controlled in the last century. However, I found several weaknesses in the manuscript, for example, presentation of the results needs to improve, the discussion of the results does not cover other continents, only being restricted to Asia and Europe. As well as several gross errors throughout the manuscript, implying that the manuscript was not revised correctly. It is necessary to improve the quality of the figures to make them self explanatory. It is necessary to improve the conclusions of the manuscript.

Response to the question of reviewer 1：Thanks for your kindly suggestion. We review the recent article and add some researches from Canada and Mexico into the discussion. We reviewed English by Editage company to avoid gross errors in our article. We also revised and improved the conclusions of the manuscript, the scientific advance was mentioned in the conclusion(L358-361, L363-365), and the control measures to reduce the occurrence of mumps among children were recommended too( L367-371).

Reviewer #2: 

The manuscript makes an interesting based on interaction between mumps and meteorological variables. It is believed that more information on the subject can still be extracted, even in face of the limitations described by authors. On this aspect, some points need to be improved throughout the text, I believe they should include the references of R software, as well as dlnm package, both described on manuscript. The introduction and materials and methods are well described. Regarding the results, I have some suggestions for improvement:

Question No.1：Replacement of Figure 1 with a boxplot of mumps cases on an annual and monthly basis.

Response to the Question No.1：Thanks for your suggestion, we replaced the figure 1 with a boxplot of mumps cases on annual and monthly basis(Fig 2.). 

Question No.2：

The other variables would be displayed in Figure 2, with monthly boxplots, making the visualization of the results much more understandable in some passages of the text, exploring the maximum and minimum.

Response to the Question No.2：Thanks for your suggestion, we replaced the figure 2 with monthly boxplots, exploring the maximum and minimum(Fig 3.).

Question No.3：

The old Figure 2 (which will be for Figure 3) could include the intensities, not only leaving the color red, this would show the interaction between them.

Response to the Question No.3：Thanks for your suggestion, we tried to change the old figure 3 with the intensities, however, we thought the original one would be better. 

Question No.4：

Improve the image resolution of all images, as they are of low quality.

Response to the Question No.4：thanks for your suggestion, we fix the image resolution of all images due to the guideline in Plos One Journal. 

Question No.5：

Paragraph preceding Figure 3, correct the pHa for hPa, on page 11. Well, from these improvements and implementations, the article will be able to be approved.

Response to the Question No.5：Thanks for your advice, We correct the pHa for hPa, on original page 11.

---

## [Decision Letter · Decision Letter 1]

23 Mar 2020

PONE-D-19-31388R1

The roles of meteorological factors on mumps incidence among children in Guangzhou, Southern China

PLOS ONE

Dear Mr Zhang,

Thank you for submitting your manuscript to PLOS ONE. After careful consideration, we feel that it has merit but does not fully meet PLOS ONE’s publication criteria as it currently stands. Therefore, we invite you to submit a revised version of the manuscript that addresses the points raised during the review process.

We would appreciate receiving your revised manuscript by May 07 2020 11:59PM. To enhance the reproducibility of your results, we recommend that if applicable you deposit your laboratory protocols in protocols.io, where a protocol can be assigned its own identifier (DOI) such that it can be cited independently in the future. For instructions see: http://journals.plos.org/plosone/s/submission-guidelines#loc-laboratory-protocols

We look forward to receiving your revised manuscript.

Kind regards,

Paulo Eduardo Teodoro, Dr.

Academic Editor

PLOS ONE

Reviewers' comments:

Reviewer's Responses to Questions

**Comments to the Author**

1. If the authors have adequately addressed your comments raised in a previous round of review and you feel that this manuscript is now acceptable for publication, you may indicate that here to bypass the “Comments to the Author” section, enter your conflict of interest statement in the “Confidential to Editor” section, and submit your "Accept" recommendation.

Reviewer #1: All comments have been addressed

Reviewer #2: All comments have been addressed

2. Is the manuscript technically sound, and do the data support the conclusions?

Reviewer #1: Yes

Reviewer #2: Yes

3. Has the statistical analysis been performed appropriately and rigorously? 

Reviewer #1: Yes

Reviewer #2: Yes

4. Have the authors made all data underlying the findings in their manuscript fully available?

Reviewer #1: Yes

Reviewer #2: No

5. Is the manuscript presented in an intelligible fashion and written in standard English?

Reviewer #1: Yes

Reviewer #2: Yes

6. Review Comments to the Author

Reviewer #1: Dear Authors.

The manuscript has improved significantly. However, some details need to be incorporated and some corrections in the figures and tables. All corrections follow from the document.

Reviewer #2: According to a new analysis carried out on manuscript about the mumps incidence in Guangzhou City - China, note a significant improvement in the writing of the text.

However, there are still some questions that can be answered:

1) Does the impact of the variables used influence similarly for each age group or sex? It is believed to be interesting information to be explored.

2) What is the main variable for the development of mumps? The results are presented throughout the text, with the data identified, so that you do not mention the most important variable in this context.

3) On line 142, create the article of the dlnm package.

4) Table 1 deserves to be better explored, because it was reproduced in the text. And based on the trend signal, which is significant, what is the interaction between meteorological variables and the mumps?

5) On line 187, an ideal word would not be correlation, but interaction.

6) On lines 204 and 205, how did you get confused. Would the RR be 60%?

7) In Figure 8, correct the word "effet" for "effect", found in the central part.

8) In Table 2, include a valve on high-effect atmospheric pressure, from 088 to 0.88.

After these changes, further details should be incorporated into the conclusions and the review. In this way, with these problems solved, the article can be accepted for a journal.

7. PLOS authors have the option to publish the peer review history of their article (what does this mean?). If published, this will include your full peer review and any attached files.

Reviewer #1: No

Reviewer #2: No

---

## [Author Response · Author response to Decision Letter 1]

8 Apr 2020

Response to Reviewers

Dear Editor,

We acknowledge with thanks the receipt of the comments of the reviewer. These comments/requirements were highly beneficial in the modification of the manuscript. The manuscript was revised according to all these comments. All modifications and/or corrections are highlighted by using the track changes mode. We would like to thank Editage (www.editage.cn) for English language editing. A point-to-point response to the reviewers’ comments was attached. We appreciate you kindly offer the opportunity to transfer our manuscript.

Reviewer #1: 

Thanks for your comments. We checked the article again and corrected the mistakes. For example, in Figure 8, we corrected the word "effet" for "effect", found in the central part.

Reviewer #2: 

Question No.1：Does the impact of the variables used influence similarly for each age group or sex? It is believed to be interesting information to be explored.

Response to the Question No.1：Thanks for your suggestion, we analyzed the cumulative extreme effects of meteorological variables on mumps cases of children by sex and age at table 3. “In the analyses by subpopulation, the extreme low effects of temperature, atmospheric pressure, and relative humidity, and the extreme high effect of wind speed showed statistical significance in the female population, but no significant extreme effect was observed in the male population. The extreme low effect of temperature was observed between 4-6 years group (RR=0.25, 95%CI: 0.11-0.60) and 15-17 years group (RR=0.01, 95%CI: 0.00-0.57). The extreme high effect of atmospheric pressure was revealed at the 4-6 years group (RR=0.46, 95%CI: 0.21-0.98)”

Question No.2：

What is the main variable for the development of mumps? The results are presented throughout the text, with the data identified, so that you do not mention the most important variable in this context.

Response to the Question No.2：

Thanks for your suggestion. We structured different model to different variables. Each variable had its own characteristic to the development of mumps. So it was hard to say which variable was more important. The strongest interaction was observed between temperature and mumps cases, and tendency was easy for the public to get. Therefore, if we much choose one of the variables as the most important factors, we will choose the temperature. 

Question No.3：

On line 142, create the article of the dlnm package. 

Response to the Question No.3：

Thanks for your suggestion, we added the reference about DLNM package into the article.

Question No.4：

Table 1 deserves to be better explored, because it was reproduced in the text. And based on the trend signal, which is significant, what is the interaction between meteorological variables and the mumps? 

Response to the Question No.4：

Thanks for your advice. We add the content to explore the table 1, “The variables of DTR, rain, pressure, wind, relative humidity were statistically significant. Wind and relative humidity had a rising tendency, whereas the DTR, rain and pressure had a decrease tendency. No tendency was observed among the variables of cases, temperature and sunshine hours.” 

Fig 5 shows the interaction network between meteorological factors and mumps cases among children, the strongest interaction was observed between temperature and mumps cases. Temperature, sunshine hour, relative correlation and aggregate rainfall had a positive interaction with mumps cases, whereas atmosphere pressure, wind speed and DTR had a negative interaction with mumps cases.

Question No.5：

On line 187, an ideal word would not be correlation, but interaction. 

Response to the Question No.5：

We change the interaction instead of the correlation on line 197, thanks for your advice.

Question No.6：

On lines 204 and 205, how did you get confused? Would the RR be 60%?

Response to the Question No.6：

Maybe the expression made confusion. I revised the article as “However, the RRs just had a significantly increasing trend before the atmosphere pressure as 992 hPa and the relative humidity as 60%, respectively.”

Question No.7：

In Figure 8, correct the word "effet" for "effect", found in the central part

Response to the Question No.7：

We are sorry for this mistake. We revised the figure 8 and correct the word “effect”.

Question No.8：

In Table 2, include a valve on high-effect atmospheric pressure, from 088 to 0.88.

Response to the Question No.8：

We are sorry for this mistake. We corrected the 088 to 0.88.

Thanks again.

---

## [Editor Report · Decision Letter 2]

13 Apr 2020

The role of meteorological factors on mumps incidence among children in Guangzhou, Southern China

PONE-D-19-31388R2

Dear Dr. Zhang,

We are pleased to inform you that your manuscript has been judged scientifically suitable for publication and will be formally accepted for publication once it complies with all outstanding technical requirements.

With kind regards,

Paulo Eduardo Teodoro, Dr.

Academic Editor

PLOS ONE

Additional Editor Comments (optional):

The authors made all the corrections requested by the reviewers and me. The manuscript has substantially improved over the first version submitted and is in a condition to be accepted.
---

## [Editor Report · Acceptance letter]

16 Apr 2020

PONE-D-19-31388R2 

The role of meteorological factors on mumps incidence among children in Guangzhou, Southern China 

Dear Dr. Zhang:

I am pleased to inform you that your manuscript has been deemed suitable for publication in PLOS ONE. Congratulations! Your manuscript is now with our production department. 

With kind regards,

on behalf of

Professor Paulo Eduardo Teodoro 

Academic Editor

PLOS ONE